# Association of Dietary Total Antioxidant Capacity with Bone Mass and Osteoporosis Risk in Korean Women: Analysis of the Korea National Health and Nutrition Examination Survey 2008–2011

**DOI:** 10.3390/nu13041149

**Published:** 2021-03-31

**Authors:** Donghyun Kim, Anna Han, Yongsoon Park

**Affiliations:** 1Department of Food and Nutrition, Hanyang University, 222 Wangsimni-ro, Seongdong-gu, Seoul 04763, Korea; apoqalik@hanyang.ac.kr; 2Department of Cancer Biology, Thomas Jefferson University, Philadelphia, PA 19107, USA; annahan8659@gmail.com

**Keywords:** bone mass, dietary total antioxidant capacity, KNHANES, menopause, osteoporosis

## Abstract

Antioxidant intake has been suggested to be associated with a reduced osteoporosis risk, but the effect of dietary total antioxidant capacity (TAC) on bone health and the risk of osteoporosis remains unclear. We aimed to assess the hypothesis that dietary TAC is positively associated with bone mass and negatively related to the risk of osteoporosis in Korean women. This cross-sectional study was performed using data from the Korea National Health and Nutrition Examination Survey. Dietary TAC was estimated using task automation and an algorithm with 24-h recall data. In total, 8230 pre- and postmenopausal women were divided into four groups according to quartiles of dietary TAC. Dietary TAC was negatively associated with the risk of osteoporosis (odds ratio, 0.73; 95% confidence interval, 0.54–0.99; *p*-value = 0.045) in postmenopausal women, but not in premenopausal women. Dietary TAC was positively associated with bone mineral content (BMC) and bone mineral density of the femoral neck and lumbar spine in postmenopausal women and BMC of the total femur and lumbar spine in premenopausal women. Our study suggests that dietary TAC is inversely associated with the risk of osteoporosis in postmenopausal women and positively associated with bone mass in both pre- and postmenopausal women.

## 1. Introduction

Osteoporosis is characterized by impairments in bone strength, elevating the risk of skeletal fractures, and is a major public health concern [1]. According to the Korea National Health and Nutrition Examination Survey (KNHANES) 2008–2011, the prevalence of osteoporosis in Korea was reported to be 17.2% in adult women and 38% in postmenopausal women (≥50 years old), indicating that menopause is an important risk factor [2]. Furthermore, the prevalence of osteoporosis in Korea has also gradually increased due to the aging population, sedentary lifestyle, and westernized dietary patterns [2,3].

Oxidative stress increases bone remodeling turnover and bone mass loss by increasing the inflammatory response and modulating the survival and differentiation of osteoblasts, whereas antioxidants inhibit oxidative stress and consequently prevent the loss of bone mass [4]. A recent study demonstrated that the intake of fruits and vegetables containing abundant antioxidants was associated with reduced bone mass loss and a reduced risk of osteoporosis in postmenopausal women [5]. Although epidemiological studies have reported a positive association between bone mineral density (BMD) and the intake of single antioxidants, such as vitamin C, vitamin E, carotenoids, and flavonoids in both premenopausal and postmenopausal women [6,7,8,9,10,11], the beneficial effects of antioxidant intake on bone health have been controversial. In fact, the consumption of vitamin C has been found to be positively correlated with the risk of osteoporosis in several studies [12,13,14], but not in others [15,16].

Interestingly, the beneficial effects of antioxidants on bone mass and the risk of osteoporosis differ depending on the anatomical site of the bones [6,8,17]. In postmenopausal women, β-carotene consumption was positively associated with femoral neck, total hip, and whole-body BMD, and the highest quintile of β-carotene consumption was associated with a decreased risk of osteopenia at the lumbar spine [6]. In contrast, increased intake of β-cryptoxanthin led to a lower risk of osteopenia at the total hip and lumbar spine, although it showed a positive correlation only with total hip BMD [6].

Previous inconsistent observations regarding the association of the intake of single antioxidants with bone mass and osteoporosis risk can be partially explained by the fact that the diet contains various antioxidants that exert cumulative and synergistic effects. Recently, dietary total antioxidant capacity (TAC) has been found to be a practical indicator to evaluate the overall antioxidant capacity of foods [18]. Rather than a simple sum of single dietary antioxidants, dietary TAC provides comprehensive information about the accumulated antioxidant capacities of different diets; therefore, dietary TAC has been applied in many studies to assess the precise preventive effects of antioxidants on diseases, including inflammatory diseases, cardiovascular diseases, and cancers [19,20,21]. However, the association between dietary TAC and the risk of osteoporosis has not yet been studied. The present study hypothesized that dietary TAC is positively associated with bone mass and negatively associated with the risk of osteoporosis in premenopausal and postmenopausal women, and analyses were performed using data from the KNHANES (2008–2011).

## 2. Materials and Methods

### 2.1. Study Population

Participants (*n* = 37,753) from the KNHANES (2008–2011) were included in this study. The KNHANES is a cross-sectional nationwide survey designed to evaluate the nutritional and health status of non-institutionalized and civilian Korean populations. The KNHANES consists of health interviews, health examinations, and nutrition surveys, and the survey uses a multistage clustered probability sampling design to select a nationally representative sample. The Institutional Review Board of the Korea National Statistical Office, Centers for Disease Control and Prevention approved the KNHANES protocol. Informed consent was obtained from all participants [22]. This study was approved by the Hanyang University Institutional Review Board (No. HYUIRB-202101-007).

The exclusion criteria for this study were as follows: male sex (*n* = 17,195); age < 19 years (*n* = 4479); missing data on bone mineral content (BMC) or BMD (*n* = 4870); missing data on baseline characteristics (*n* = 1395); Z-scores of dietary TAC, BMD, and BMC greater than +3.29 or less than −3.29 (*n* = 251); extreme energy intake (<500 kcal/day or >4000 kcal/day; *n* = 250); renal failure (*n* = 19); pregnancy or lactation (*n* = 148); hormone replacement therapy (*n* = 684); and osteoporosis treatment (*n* = 232). A total of 8230 female participants were included in the final analyses.

### 2.2. Study Variables

A health interview questionnaire was used to obtain data on age, sex, postmenopausal status, smoking status, alcohol consumption, and exercise status. Postmenopausal status was defined on the basis of a self-reported questionnaire survey containing questions regarding whether one year had passed since the last menstruation or hysterectomy. Waist circumference (WC) and body mass index (BMI) were measured as indices of body measurements. Regular exercise was defined as walking for more than 30 min at least five times per week or vigorous physical activity for at least 20 min per day, three days per week. For biochemical analysis, a fasting blood sample was taken, refrigerated, and analyzed within 24 h at the Central Testing Center in Seoul, Korea. The serum concentration of 25 hydroxyvitamin D (25(OH)D) was measured using the radioimmunoassay method (1470 WIZARD Gamma-Counter, PerkinElmer, Finland).

Dietary intake data were collected using one day of the 24-h recall method during household interviews. Daily intakes of energy, protein, calcium, phosphorus, potassium, and sodium were calculated using the Korean Food Composition Table [23].

BMC and BMD at the lumbar spine (L1–L4), total femur, and femoral neck were measured by trained technicians using dual-energy X-ray absorptiometry (DISCOVERY-W fan-beam densitometer; HOLOGIC, MA, USA). Osteopenia or osteoporosis was diagnosed using the minimum T-score of BMD in three sites (lumbar spine, total femur, and femoral neck) based on the criteria of the KNHANES [24]. Briefly, the T-scores of the lumbar spine, total femur, and femoral neck were classified as follows: greater than or equal to −1, normal; less than −1 to more than −2.5, osteopenia; and less than or equal to −2.5, osteoporosis.

### 2.3. Development of the TAC Database and Estimation of Dietary TAC

According to Floegel et al. [18], the theoretical TAC value of each food item was obtained by summing the TAC values of individual antioxidants derived by multiplying the contents of 42 individual antioxidants in the food by their antioxidant capacities; antioxidant capacities were determined by the 2,2-azino-bis-3-ethylbenzthiazoline-6-sulphonic acid assay and expressed as vitamin C equivalents (VCE). The individual antioxidants were β-carotene, α-carotene, β-cryptoxanthin, lycopene, lutein, zeaxanthin, retinol, ascorbic acid, α-tocopherol, γ-tocopherol, quercetin, kaempferol, myricetin, isorhamnetin, luteolin, apigenin, hesperetin, naringenin, eriodictyol, (+)-catechin, (+)-gallocatechin, (–)-epicatechin, (–)-epigallocatechin, (–)-epicatechin 3-gallate, (–)-epigallocatechin 3-gallate, theaflavin, theaflavin 3-gallate, theaflavin 3′-gallate, theaflavin 3,3′-digallate, cyanidin, delphinidin, malvidin, pelargonidin, peonidin, petunidin, daidzein, genistein, glycitein, biochanin A, formononetin, procyanidin, and cinnamtannin B1.

The contents of 42 individual antioxidants were collected from databases produced by the Korea Rural Development Administration (RDA) [25,26] and the United States Department of Agriculture [27,28,29]. When two or more values for an individual food item were available from these databases, the RDA value was prioritized. For food items without available values in these government publications, we used the values from published papers [30,31,32,33]. The database was expanded by estimating values based on similar food items and moisture conversion factors suggested in the KNHANES, and by applying logical zeros. The missing values were denoted as zeros [34]. The database covered 95.34% of the food intake.

Dietary TAC was estimated by creating a program linking food consumption data from the KNHANES with the TAC database. The dietary TAC (mg VCE) of individual food items was determined by multiplying the daily consumption of the item (g) by the theoretical TAC of each food (mg VCE/100 g) in our TAC database (Equation (1)). Daily TAC from the diet was the sum of daily TAC from all food items reported in the one day of 24-h recall (Equation (2)). This series of operations was performed efficiently through computational automation (Appendix A).

Equation (1):(1)$$Theoretical\;TAC=\sum \left(antioxidant\;content\frac{mg}{100g}\times antioxidant\;capacity\frac{mg\;VCE}{100g}\right)$$

Equation (2):(2)Daily dietary TAC=∑(Daily intake food amount × theoretical TAC (per 1g of food ))

### 2.4. Statistical Analyses

All statistical analyses were performed using complex sample survey data in SPSS version 26.0 (SPSS Inc., Chicago, IL, USA). Sample weights obtained from the KNHANES were used to obtain unbiased estimates of means and frequencies that were nationally representative of the Korean population [35]. Continuous variables were expressed as the mean ± standard error of the mean, and categorical variables were expressed as frequencies and percentages. The characteristics of and risk factors for osteoporosis were compared between the groups using the Student’s *t*-test for continuous variables and the chi-square test for categorical variables. The values of dietary TAC, BMC, or BMD at the total femur, femoral neck, and lumbar spine were standardized to identify univariate outliers.

Multiple regression analyses were performed to screen for unsuitable potential covariates and to assess the relationships between dietary TAC and BMC or BMD, considering the suitable potential covariates. In multivariate models, covariates showing a *p*-value < 0.20 were selected as confounding factors and included in the adjusted model [36]. All groups were subdivided into four groups according to quartiles of dietary TAC. Analysis of covariance (ANCOVA) with Bonferroni correction was performed to assess mean differences in BMC or BMD among the quartiles after adjustment for confounding variables. A multivariable logistic regression model was used to examine the association between dietary TAC and the risk of osteoporosis. The *p*-value for the trend was calculated by employing multivariate logistic regression analyses and by handling the median value of each dietary TAC quartile as a continuous value. Statistical significance was set at a *p*-value < 0.05.

## 3. Results

### 3.1. Study Design and Baseline Characteristics of Participants

Figure 1 outlines the study design and selection of participants for the present study. Table 1 presents the participants’ baseline characteristics and the risk factors of osteoporosis in the study population. Both postmenopausal and premenopausal women with osteoporosis had a significantly lower WC, BMI, BMC, and BMD than those without osteoporosis. In postmenopausal women, participants with osteoporosis were older, had a higher smoking frequency, had lower alcohol consumption, and engaged in more regular patterns of exercise than those without osteoporosis; additionally, they also had lower dietary intakes of energy, protein, calcium, phosphorous, potassium, and sodium than those without osteoporosis. Premenopausal women with osteoporosis had markedly lower 25(OH)D levels than those without osteoporosis.

### 3.2. Associations between Dietary TAC and Bone Health

Logistic regression analysis showed that dietary TAC was negatively associated with the risk of osteoporosis before and after adjustment for potential confounders in postmenopausal women, but not in premenopausal women (Table 2). Postmenopausal women consuming food items corresponding to a daily dietary TAC of ≥456.89 mg VCE/day had a low risk of osteoporosis. Furthermore, there were significant positive correlations between dietary TAC, BMC, and BMD in postmenopausal women (Table 3). Dietary TAC was positively correlated only with the BMC of the total femur and lumbar spine in premenopausal women.

Lastly, ANCOVA showed that dietary TAC was positively associated with the BMC of the lumbar spine, total femur, and femoral neck and the BMD of the lumbar spine and femoral neck before and after adjustment for potential confounders in postmenopausal women (Table 4). In premenopausal women, there was a significant and positive association between dietary TAC and the BMC of the total femur and lumbar spine (Table 4).

## 4. Discussion

The present study showed that dietary TAC was positively associated with BMC and BMD and inversely associated with the risk of osteoporosis in postmenopausal women. This study also found a positive correlation between dietary TAC and the BMC of the lumbar spine and total femur in premenopausal women.

The intake of fruits and vegetables has a beneficial effect on bone mass in postmenopausal women [4,37]. Antioxidants present in fruits and vegetables have been highlighted as one of the key causes of beneficial outcomes for bone health [5]. Epidemiological studies have shown that BMD is positively related to the intake of vitamin C [38], vitamin E [8], carotenoids [6], and flavonoids [10,11] in postmenopausal women. Collectively, these observations indicated that the BMD of the femoral neck was strongly affected by the intake of antioxidants, while the BMD of the total hip and lumbar spine was slightly affected in postmenopausal women. In the present study, dietary TAC was significantly associated with the BMD of the femoral neck and lumbar spine, but not with the BMD of the total femur in postmenopausal women. It has been reported that the accumulation of yellow bone marrow due to aging is known to exert an effect related to an increase of inflammatory substances and inhibition of bone formation, and, as a result, inhibit skeletal homeostasis and cause a reduction in BMD [39,40], particularly femur BMD [41]. Thus, the confounding effects of variables such as age must be considered for the precise assessment of the effect of dietary TAC on femur BMD in postmenopausal women.

Furthermore, Li et al. [37] reported that the intake of fruits and vegetables was significantly associated with whole-body BMC, but not with the BMC and BMD of specific sites in premenopausal women. In premenopausal women, vitamin C intake was positively associated with femur BMD [42], vitamin E intake was positively correlated with lumbar BMD [9], and carotenoid intake was positively linked to the BMD of the total hip [6]. These findings suggest that the effect of a single antioxidant on BMD could be site-specific in premenopausal women. The present study also showed that dietary TAC was positively associated with the BMC of the lumbar spine and total femur in premenopausal women. In premenopausal women, bone loss caused by oxidative stress could be more prominent in the trabecular bone than in the cortical bone [5], therefore, sites with a greater proportion of trabecular bone, such as the lumbar spine and total femur, were more sensitive to antioxidant effects than the femoral neck [43].

The inverse association between the intake of fruit and vegetables and the risk of osteoporosis in postmenopausal women has been demonstrated previously [4,5,44], although some studies have shown controversial results regarding the association between the intake of single antioxidants and osteoporosis risk. Cross-sectional studies showed a significant negative association between the risk of osteoporosis and vitamin C intake in postmenopausal women (*n* = 1196 [12] and *n* = 1878 [13]) and men and women aged ≥ 50 years (*n* = 3047) [14]. In contrast, a cohort study reported that vitamin C and vitamin E intake was not related to the risk of osteoporosis in postmenopausal women (*n* = 187) [16]; a case-control study also reported that vitamin C intake was not associated with osteoporosis risk in postmenopausal women (*n* = 144) [15]. The small sample sizes in these studies might have reduced the accuracy of the statistical analyses [45], implying that the inconsistencies in the above findings might be due to small sample sizes. Our study analyzed a relatively large sample population (*n* = 3559) and demonstrated an inverse association between dietary TAC and the risk of osteoporosis in postmenopausal women.

The correlation between the risk of osteoporosis and the intake of fruits, vegetables, or antioxidants in premenopausal women has been highlighted and studied to a lesser extent, because estrogen prevents bone loss by maintaining bone homeostasis and lowering the risk of osteoporosis in premenopausal women [46]. The present study found no association between the risk of osteoporosis and dietary TAC in premenopausal women that might result from a very low incidence of osteoporosis in premenopausal women (1.0–1.7%) in the current study. 

Independent with BMD, several biomarkers have been addressed to evaluate bone health and the risk of osteoporosis and osteoporotic fractures. Advanced glycation end products (AGEs), such as pentosidine, were associated with vertebral fracture, but not with lumbar BMD. [47]. Imbalance of serum homocysteine (Hcy) level has also been reported as a pathological biomarker of osteoporotic fractures [48]. Thus, it will be interesting to study the effects of a high dietary TAC diet on serum AGEs and Hcy levels, as well as the risk of osteoporotic fractures.

Consistent with our results, a vegan diet has been suggested to be associated with better bone health compared to non-vegans due to the inverse correlation between fruit and vegetable consumption and bone health [5,37]. However, a recent study found that vegetarians and vegans showed a lower BMD at the femoral neck and lumbar spine compared to non-vegans [49], which might come from the insufficient intake of dietary calcium, vitamin D, and high biological value proteins in the vegan diet [50]. The present study also observed that postmenopausal women with osteoporosis had a lower intake of calcium, phosphorous, protein, and energy intake, and premenopausal women with osteoporosis had a reduced level of 25(OH)D levels than those without osteoporosis. Since the original survey was a cross-sectional study, a cause-and-effect association between the lower nutrient levels and the risk of osteoporosis was unclear. However, high dietary TAC was still significantly inversely associated with the risk of osteoporosis in postmenopausal women and positively correlated to the site-specific BMC and BMD in both pre-and postmenopausal women after adjusting for energy intake. These observations suggest that dietary TAC might also need to be considered for the evaluation of the inverse correlation of a vegan diet and the risk of osteoporosis and bone health in future studies.

The beneficial association between antioxidant intake, bone mass, and the risk of osteoporosis has been studied in many countries, including the Republic of Korea [6,8,12,14,15], China [10,13], and Japan [9]. However, these studies primarily focused on the effects of single antioxidant intake, which might not fully explain the effects of all antioxidants in the diet. By calculating the TAC in the diet, this study evaluated the cumulative and overall antioxidant effects on bone mass and the risk of osteoporosis in the Korean female population.

To our knowledge, this is the first study to examine the association between dietary TAC and bone health in women with a relatively large sample size. The power verification had a value of more than 0.8, and the hypothesis verification of this study could be trusted. Rather than evaluating the effects of single antioxidants, the overall antioxidant capacity of food items was evaluated, and the accuracy of the assessments was improved. The limitations of this study include the following: (1) a cause-and-effect relationship between dietary TAC and the risk of osteoporosis could not be established due to the cross-sectional design of the original survey, (2) there is a possibility of potential residual confounders, and (3) there is a potential of assessment errors regarding food intake, since the one day of dietary records was based on participants’ memory.

## 5. Conclusions

The present study showed that dietary TAC was positively associated with bone mass at the lumbar spine and femoral neck and inversely associated with the risk of osteoporosis in postmenopausal women. In premenopausal women, dietary TAC was positively associated with the BMC of the lumbar spine and total femur. These observations indicate that the consumption of foods with high dietary TAC, such as grapes, radish leaves, pepper paste, oranges, and spinach, could improve bone health, particularly in postmenopausal women. Additional long-term cohort or intervention studies are needed to confirm the effect of a diet with high dietary TAC on bone health and the risk of osteoporosis.

## Figures and Tables

**Figure 1 nutrients-13-01149-f001:**
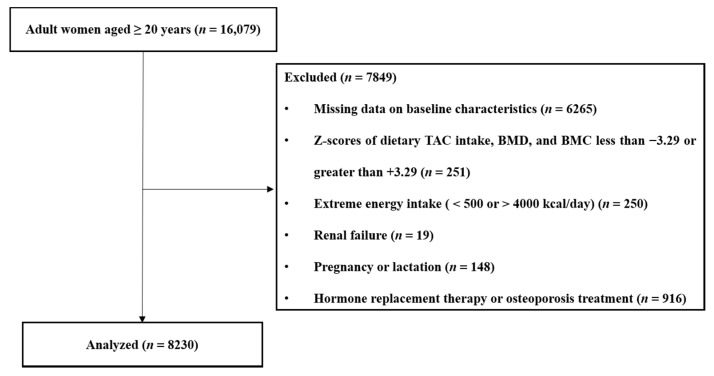
Summary of participant selection in the present study. TAC, total antioxidant capacity; BMC, bone mineral content; BMD, bone mineral density.

**Table 1 nutrients-13-01149-t001:** Baseline characteristics of the participants and risk factors of osteoporosis in the study population.

Variables	Postmenopausal Women		*p*-Value	Premenopausal Women		*p*-Value
	Non-Osteoporosis (*n* = 2260)	Osteoporosis (*n* = 1299)		Non-Osteoporosis (*n* = 4606)	Osteoporosis (*n* = 65)	
Age (years)	60.46 ± 0.19	69.27 ± 0.22	<0.001	36.70 ± 0.13	37.69 ± 1.14	0.670
WC (cm)	83.82 ± 0.19	80.77 ± 0.27	<0.001	75.34 ± 0.14	68.80 ± 0.81	<0.001
BMI (kg/m^2^)	24.81 ± 0.07	23.28 ± 0.09	<0.001	22.65 ± 0.05	19.98 ± 0.30	<0.001
Smoking status, *n* (%)			0.002			0.443
Never	2100 (93.7)	1,149 (89.0)		4122 (89.8)	57 (87.7)	
Former	56 (2.5)	61 (4.7)		184 (4.0)	1 (1.5)	
Current	86 (3.8)	81 (6.3)		285 (6.2)	7 (10.8)	
Alcohol consumption, *n* (%)	627 (28.0)	246 (19.1)	<0.001	2330 (50.8)	26 (41.3)	0.162
Regular exercise, *n* (%)	557 (24.9)	219 (17.0)	<0.001	1034 (22.6)	13 (20.0)	0.367
25(OH)D (ng/mL)	18.66 ± 0.15	18.42 ± 0.20	0.116	16.05 ± 0.08	14.11 ± 0.73	0.047
Dietary intake						
Energy intake (kcal/day)	1587.56 ± 11.43	1449.32 ± 13.78	<0.001	1694.50 ± 8.77	1686.48 ± 67.41	0.517
Protein (g/day)	53.81 ± 0.551	45.87 ± 0.663	<0.001	61.93 ± 0.411	62.19 ± 3.126	0.915
Calcium (mg/day)	435.70 ± 6.43	357.66 ± 12.24	<0.001	457.82 ± 4.15	474.80 ± 32.74	0.491
Phosphorus (mg/day)	974.54 ± 8.54	851.73 ± 10.22	<0.001	1025.90 ± 5.96	1067.54 ± 47.65	0.472
Potassium (mg/day)	2695.38 ± 29.90	2245.41 ± 35.53	<0.001	2728.75 ± 18.57	2826.00 ± 162.31	0.468
Sodium (mg/day)	3913.31 ± 53.89	3513.67 ± 70.59	<0.001	4344.46 ± 42.51	4773.58 ± 384.46	0.302
BMC (g)						
Lumbar spine	48.87 ± 0.233	34.73 ± 0.239	<0.001	55.92 ± 0.155	41.27 ± 0.729	<0.001
Total femur	28.22 ± 0.092	22.76 ± 0.104	<0.001	29.21 ± 0.064	21.95 ± 0.387	<0.001
Femoral neck	3.34 ± 0.010	2.59 ± 0.011	<0.001	3.61 ± 0.008	2.70 ± 0.041	<0.001
BMD (g/cm^2^)						
Lumbar spine	0.87 ± 0.002	0.67 ± 0.003	<0.001	0.95 ± 0.002	0.74 ± 0.010	<0.001
Total femur	0.83 ± 0.002	0.67 ± 0.002	<0.001	0.88 ± 0.002	0.70 ± 0.009	<0.001
Femoral neck	0.68 ± 0.002	0.53 ± 0.002	<0.001	0.74 ± 0.001	0.56 ± 0.007	<0.001

Values are expressed as the mean ± standard error of the mean for continuous variables or as the number (percentage) for categorical variables. WC, waist circumference; BMI, body mass index; BMC, bone mineral content; BMD, bone mineral density.

**Table 2 nutrients-13-01149-t002:** Associations between dietary total antioxidant capacity (TAC) and the risk of osteoporosis in the study population.

Variables	Quartiles of Dietary TAC (mg VCE/day)				*p*-Value
	Q1	Q2	Q3	Q4	
Postmenopausal women, *n* (osteoporosis %)	890 (47.0)	890 (37.5)	890 (33.4)	889 (28.1)	
Cut off (mg VCE/day)	<150.96	150.96 ≤ to < 267.65	267.65 ≤ to < 456.89	≥456.89	
Median dietary TAC (mg VCE/day)	93.48	208.26	347.32	643.52	
Unadjusted OR (95% CI)	1	0.673 (0.525–0.863)	0.618 (0.486–0.787)	0.398 (0.315–0.504)	<0.001
Adjusted OR (95% CI)	1	0.952 (0.715–0.268)	1.066 (0.805–1.410)	0.732 (0.540–0.992)	0.045
Premenopausal women, *n* (osteoporosis %)	1167 (1.7)	1168 (1.0)	1168 (1.4)	1168 (1.5)	
Cut off (mg VCE/day)	<169.28	169.28 ≤ to < 286.79	286.79 ≤ to < 474.48	≥474.48	
Median dietary TAC (mg VCE/day)	109.36	226.41	361.06	666.44	
Unadjusted OR (95% CI)	1	0.515 (0.216–1.228)	0.708 (0.337–1.487)	1.056 (0.489–2.283)	0.590
Adjusted OR (95% CI)	1	0.554 (0.232–1.318)	0.825 (0.395–1.723)	1.220 (0.555–2.681)	0.394

Odds ratios (ORs) and 95% confidence intervals (CIs) are presented. The logistic regression model was adjusted for age, waist circumference, energy intake, sodium intake, smoking status, and regular exercise for postmenopausal women, and adjusted for waist circumference and 25(OH)D levels for premenopausal women. VCE, vitamin C equivalents.

**Table 3 nutrients-13-01149-t003:** Correlation of dietary total antioxidant capacity (TAC) with bone mass.

Variables	Postmenopausal Women		Premenopausal Women	
	*r* ^1^	*p*-Value	*r* ^1^	*p*-Value
BMC (g)				
Lumbar spine	0.146	<0.001	0.043	0.005
Total femur	0.153	<0.001	0.034	0.037
Femoral neck	0.179	<0.001	0.008	0.646
BMD (g/cm^2^)				
Lumbar spine	0.144	<0.001	0.023	0.154
Total femur	0.170	<0.001	0.014	0.395
Femoral neck	0.177	<0.001	0.012	0.480

^1^ All values represent correlations (r). BMC, bone mineral content; BMD, bone mineral density.

**Table 4 nutrients-13-01149-t004:** Bone mass according to quartiles of dietary total antioxidant capacity (TAC) in the study population.

Variables	Quartiles of Dietary TAC (mg VCE/day)				*p*-Value ^a^	*p*-Value ^b^
	Q1	Q2	Q3	Q4		
Postmenopausal women, *n*	890	890	890	889		
Dietary TAC (mg VCE/day)	90.10 ± 1.32	207.92 ± 1.14	352.61 ± 1.81	726.45 ± 8.54		
BMC (g)						
Lumbar spine	40.86 ± 0.409	43.50 ± 0.421	44.07 ± 0.411	45.94 ± 0.409	<0.001	0.049
Total femur	25.11 ± 0.166	26.01 ± 0.163	26.61 ± 0.158	27.19 ± 0.166	<0.001	0.030
Femoral neck	2.89 ± 0.019	3.05 ± 0.019	3.11 ± 0.019	3.21 ± 0.019	<0.001	0.016
BMD (g/cm^2^)						
Lumbar spine	0.77 ± 0.005	0.80 ± 0.005	0.81 ± 0.005	0.82 ± 0.005	<0.001	0.017
Total femur	0.74 ± 0.004	0.77 ± 0.004	0.78 ± 0.004	0.80 ± 0.004	<0.001	0.080
Femoral neck	0.59 ± 0.004	0.62 ± 0.004	0.63 ± 0.004	0.65 ± 0.004	<0.001	0.025
Premenopausal women, *n*	1167	1168	1168	1168		
Dietary TAC (mg VCE/day)	105.70 ± 1.16	226.70 ± 1.00	369.13 ± 1.60	762.04 ± 8.00		
BMC (g)						
Lumbar spine	58.42 ± 0.283	59.10 ± 0.283	59.49 ± 0.282	60.03 ± 0.297	0.001	0.002
Total femur	29.21 ± 0.137	29.44 ± 0.139	29.85 ± 0.142	29.92 ± 0.141	0.006	0.015
Femoral neck	3.72 ± 0.017	3.71 ± 0.016	3.76 ± 0.016	3.76 ± 0.017	0.527	0.850
BMD (g/cm^2^)						
Lumbar spine	0.98 ± 0.003	0.99 ± 0.003	0.99 ± 0.003	0.99 ± 0.004	0.029	0.050
Total femur	0.90 ± 0.003	0.90 ± 0.003	0.91 ± 0.003	0.91 ± 0.003	0.093	0.231
Femoral neck	0.76 ± 0.003	0.76 ± 0.003	0.77 ± 0.003	0.77 ± 0.003	0.849	0.856

All values are presented as the mean ± standard error of the mean or the number of participants, as appropriate. ^a^ Unadjusted p-value for the differences in BMC and BMD according to quartiles of dietary TAC; ^b^ p-value for the differences in BMC and BMD according to quartiles of dietary TAC after adjustment for confounding factors. The covariates were age, waist circumference, energy intake, sodium intake, smoking status, and regular exercise for postmenopausal women, and waist circumference and 25(OH)D levels for premenopausal women. BMC, bone mineral content; BMD, bone mineral density.

## Data Availability

Data from the Korea National Health and Nutrition Examination Survey (KNHANES) 2008–2011 can be found on the official website of the KNHANES (http://knhanes.cdc.go.kr, accessed on 30 March 2021).

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
