# Peer review of "Association of Dietary Total Antioxidant Capacity with Bone Mass and Osteoporosis Risk in Korean Women: Analysis of the Korea National Health and Nutrition Examination Survey 2008–2011"

_nutrients, 2021, doi:10.3390/nu13041149_

Round 1

Reviewer 1 Report

Congratulation to the authors for the nice work conducted

INTRODUCTION

The introduction provides sufficient background information for readers to understand the research problem.

Include a concise study objective.

METHODS

The experimental approach is appropriate for the aim of the study.

This section is well described and allows to replicate the study.

RESULTS

Results paragraphs include more relevant and extended data.

All of the tables include specific, well-developed statistic.

DISCUSSION

The interpretations of the data considered are consistent.

Explain the implications of results obtained and their practical applications

LITERATURE CITED

The literature cited is relevant to the study.

SIGNIFICANCE AND NOVELTY

As it stands, the results are novel and important enough for this journal.

Author Response

Response to Reviewer 1 Comments

Major Points

Point 1: Congratulation to the authors for the nice work conducted. The introduction provides sufficient background information for readers to understand the research problem and it includes a concise study objective. The experimental approach is appropriate for the aim of the study and method section is well described and allows to replicate the study. Results paragraphs include more relevant and extended data, and all the tables include specific, well-developed statistic. The interpretations of the data considered are consistent and discussion explains the implications of results obtained and their practical applications. As it stands, the results are novel and important enough for this journal.

Response 1: Thank you for your positive comment. We have listed several food sources containing high dietary TAC in line: 314-315.

Reviewer 2 Report

The review of the manuscript entitled on Association of dietary total antioxidant capacity with bone mass and osteoporosis risk in Korean women: Analysis of the Korean National Health and Nutritional Examination Survey 2008-2011.

Firstly, the aim of the study indicated a potential new pathogenesis of osteoporosis. However, the examination of the present study was very limited, because the authors focused on the role of antioxidant only on bone mineral density. This focusing inhibits to spread the possible pathogenesis of oxidation of the tissue. Furthermore, if the dietary TAC intake plays an important role on the maintenance of bone mineral density, the bone health in the people who take vegan diet may have better bone health comparing to that in non-vegans. However, the recent meta-analysis (Iguacel I et al Nutrition Rev 77: 1-18: 2019.) clearly indicated that the BMD in vegans were less than the BMD in non-vegans. This review report seemed to be opposite to your observation. The reviewer suggested that the authors would like to discuss about the difference in food intake difference between your high TAC group and vegans.

Major criticism

  • The dietary TAC was calculated by using one day recall. The reference number 18 may indicate the accuracy and precision of TAC calculation. However, the authors indicate the precision and accuracy of the TAC in your system, because whether the subject’s one day recall can be representative for the subject’s intake of TAC during life span or not.
  • The diagnosis of osteoporosis was made by BMD only. However, the diagnosis of osteoporosis should be made by the presence or absence of prevalent fragility fractures and BMD. The authors indicate the presence of fragility fractures in the subjects. This is very important point because osteoporosis may consist two different types, one has low BMD and the other part of osteoporosis with fracture without low BMD. The latter part of osteoporosis has been considered to have deteriorated mechanical properties of bone tissue through tissue oxidation or glycation. Very recently Nakamura Y et al reported the association of fractures with pentosidine or CML (Scientific Reports https://doi.org/10.1038/s41598-020-78993-w.) The former glycated AGE was associated with fractures regardless of BMD and the latter AGE associated with low BMD. Serum level of homocysteine, which is an enhancer of tissue oxidative stress, has been reported as a cause of fracture independently from BMD. Therefore, daily TAC intake may prohibit tissue accumulation of oxidants or AGE. This is a reason why the reviewer wants to know the fracture presence in the participants.

Your cohort may not have a pure vegan’s diet even the people with highest quartile of TAC. Thus, the reviewer wants to know the other dietary components which may contribute to sustain higher BMD after menopause.  

Reviewer 3 Report

Please see attached for minor requested revisions

Round 2

Reviewer 2 Report

The revised manuscript properly modified and is acceptable for publication in Nutrients. Therefore, no additional comments is raised by the reviewer.